# EPISTEMIC INTEGRITY IN LARGE LANGUAGE MODELS

**Bijean Ghafouri**[*,1]  **Shahrad Mohammadzadeh**[*,2,7]  **James Zhou**[3]

**Pratheeksha Nair**[2,7]  **Jacob-Junqi Tian**[4,7]  **Mayank Goel**[5]

**Reihaneh Rabbany**[2,7]  **Jean-François Godbout**[6,7]  **Kellin Pelrine**[2,7]

[1]University of Southern California  [2]McGill University  [3]UC Berkeley
[4]Vector Institute  [5]IIIT Hyderabad  [6]Université de Montréal  [7]Mila
 **Correspondence:** kellin.pelrine@mila.quebec

## ABSTRACT

Large language models are increasingly relied upon as sources of information, but their propensity for generating false or misleading statements with high confidence poses risks for users and society. In this paper, we confront the critical problem of epistemic miscalibration—where a model's linguistic assertiveness fails to reflect its true internal certainty. We introduce a new human-labeled dataset and a novel method for measuring the linguistic assertiveness of Large Language Models which cuts error rates by over 50% relative to previous benchmarks. Validated across multiple datasets, our method reveals a stark misalignment between how confidently models linguistically present information and their actual accuracy. Further human evaluations confirm the severity of this miscalibration. This evidence underscores the urgent risk of the overstated certainty Large Language Models hold which may mislead users on a massive scale. Our framework provides a crucial step forward in diagnosing and correcting this miscalibration, offering a path to safer and more trustworthy AI across domains.

## 1 INTRODUCTION

Large Language Models (LLMs) have markedly transformed how humans seek and consume information, becoming integral across diverse fields such as public health (Ali et al., 2023), coding (Zambrano et al., 2023), and education (Whalen & et al., 2023). Despite their growing influence, LLMs are not without shortcomings. One notable issue is the potential for generating responses that, while convincing, may be inaccurate or nonsensical—a long-standing phenomenon often referred to as "hallucinations" (Jo, 2023; Huang et al., 2023; Zhou et al., 2024b).

A critical aspect of trustworthiness in LLMs is *epistemic calibration*—the alignment between a model's internal confidence in its outputs and the way it expresses that confidence through natural language. Misalignment between internal certainty and external expression can lead to users being misled by overconfident or underconfident statements, posing significant risks in high-stakes domains such as legal advice, medical diagnosis, and misinformation detection. While of great normative concern, how LLMs express linguistic uncertainty has received relatively little attention to date (Sileo & Moens, 2023; Belem et al., 2024).

Figures 4 and 5 in section A of the Appendix illustrate the issue of epistemic calibration providing insights into the operation of certainty in the context of human interactions with LLMs. We highlight the following key points in these figures:

- **Distinct Roles of Certainty:** Internal certainty and linguistic assertiveness have distinct functions within LLM interactions, underlining their unique contributions to how they shape individual beliefs.

---

*Equal contribution.

- **Human access to LLM certainty:** Linguistic assertiveness holds a critical role as the primary form of certainty available to users. Unlike internal certainty, which remains hidden within the model's computational processes, linguistic assertiveness is directly perceivable and influences how users interpret the model's outputs.
- **Beyond Content:** Users retrieve more than just the content from an LLM's output. The style and assertiveness of the language used also play a significant role, shaping perceptions through the communication of certainty. This interaction between the model's output and its linguistic assertiveness is crucial for understanding the full impact on individual perceptions.

Several studies have explored the calibration of internal confidence in LLMs. For instance, Zhang et al. (2024) examine confidence calibration, proposing techniques to reduce hallucinations and enhance the model's ability to answer known questions while avoiding unknown ones. However, they overlook the role of *linguistic assertiveness* and how external certainty can still lead to epistemic miscalibration even if internal confidence is addressed. Similarly, Ren et al. (2023) focus on factual knowledge and LLM behavior before and after retrieval-augmented generation (RAG). While they investigate internal confidence, they fail to frame miscalibration as an end-to-end issue involving both internal certainty and linguistic assertiveness, ignoring the interplay between model predictions and how confidence is expressed linguistically.

More recent studies aim to bridge the gap between internal confidence and linguistic assertiveness but still face considerable limitations. Mielke et al. (2022) explore epistemic calibration but use limited scoring scales to measure both assertiveness and confidence, restricting continuous assessments. Their models also rely on a narrow range of datasets, which limits their applicability across diverse domains. Zhou et al. (2024a) address miscalibration using epistemic markers, but their method lacks real domain grounding and fails to consider the complexities of language. In contrast, our study overcomes these limitations by fine-tuning models for a broader scope of topics and domains, achieving state-of-the-art performance in assertiveness estimation.

This review of existing work on LLM calibration and confidence reveals several gaps that our research aims to address:

- **Lack of Integrated Approaches**: Previous studies address either internal certainty or linguistic assertiveness but rarely both simultaneously (Jiang et al., 2021). There is a need for comprehensive frameworks that integrate these aspects to ensure LLMs communicate accurately and responsibly.
- **Inadequate Assertiveness Measurement**: Existing methods for measuring linguistic assertiveness rely heavily on lexicon-based approaches (Pei & Jurgens, 2021) or subjective perceptions without adequate validation (Steyvers et al., 2024). These methods often lack contextual depth and fail to generalize across diverse domains.
- **Limited High-Stakes Evaluation**: Although some studies explore epistemic calibration, they cover a narrow range of topics and employ low-resolution measures of assertiveness, limiting their applicability in critical domains such as misinformation detection (Mielke et al., 2022).

To address these gaps, our paper provides:

- **A New Assertiveness Detection Model**: We train a new model using a diverse dataset to detect linguistic assertiveness, improving the accuracy relative to previous approaches by incorporating contextual nuance and aligning more closely with human perceptions. This approach also addresses limitations in lexicon-based methods, enhancing generalizability across domains.
- **Empirical Evidence of Epistemic Miscalibration**: Our work provides a comprehensive comparison between internal certainty and linguistic assertiveness, documenting instances of miscalibration across a broad range of domains. Our experiments reveal that LLMs frequently generate highly assertive explanations despite low internal certainty, which can mislead users.
- **Validation with Human Perception**: We conduct comprehensive surveys assessing human perceptions of LLMs' linguistic assertiveness, confirming that misalignment poses a real risk. This addresses the gap regarding alignment of computational measures with subjective human perceptions of language.

This study also presents a novel human-centered approach for developing a robust assertiveness scoring method, introducing a new dataset from multiple domains. To ensure reliability, we train and compare several models, identifying the best estimator for assertiveness based on accuracy and transferability across different contexts. After selecting the top-performing model, we validate

the results using human-surveyed data from the LIAR dataset (Wang, 2017), which was coded by participants different from those who annotated the primary dataset. This comprehensive methodology enables us to thoroughly assess both the objective and subjective aspects of assertiveness in language model explanations.

Our findings reveal that when the model has low internal certainty, it generates explanations that are significantly over-assertive, meaning the language used implies a higher degree of certainty than warranted by the model's actual confidence or knowledge base. This miscalibration could lead users to misconstrue the model's judgments as more reliable than they actually are. Our results confirm a strong correlation between the GPT 4o's assertiveness scores and human perceptions of assertiveness, but a weak correlation between human perceptions and internal certainty, and an even weaker relationship between GPT 4o model assertiveness and internal certainty.[1]

## 2 CONCEPTUALLY UNDERSTANDING CERTAINTY AND ASSERTIVENESS IN NATURAL LANGUAGE

### 2.1 CERTAINTY

To elucidate the challenges of epistemic calibration, it is essential to understand the concepts of certainty and assertiveness in natural language communication. These foundational notions underpin how information is conveyed by LLMs and interpreted by users.

Effective communication hinges on the accurate conveyance of certainty, enabling individuals and systems to assess the reliability of information. In human communication, speakers use linguistic cues to express their confidence levels, which listeners interpret to form judgments about the truthfulness and credibility of statements (Budescu & Wallsten, 1985; Clarke et al., 1992). Similarly, for LLMs, effectively conveying certainty is crucial to ensure users can trust and interpret the provided information accurately. In this section, we decompose *certainty* into two key concepts: **Internal Certainty** and **External Certainty**. Misalignment between these two dimensions can potentially lead to the need for **Epistemic Calibration**, as discussed in Section 2.2.

#### 2.1.1 INTERNAL CERTAINTY

Internal certainty, also referred to as model confidence, represents the probability an LLM assigns to a particular output based on its internal computations and parametric knowledge from its training data. In tasks such as question-answering, internal certainty is often represented by the probability the model assigns to its selected response compared to alternative answers (Jiang et al., 2021; Hendrycks et al., 2021).

**Internal Confidence Estimation**    Hendrycks et al. (2021) introduce baseline methods to detect misclassifications by aligning model confidence with the true probability of correctness. Token-level analysis is a common approach to generate fine-grained uncertainty estimates by examining probabilities assigned to individual tokens (Jiang et al., 2021; Kuhn et al., 2023; Duan et al., 2024). Another method assesses variability across multiple outputs, where higher variability indicates greater uncertainty (Xiong et al., 2024). External classifiers can also predict uncertainty by analyzing both input data and the model's internal representations, offering a more comprehensive evaluation (Shrivastava et al., 2023). Recent work also focuses on making internal confidence more interpretable, using both numerical scores (Lin et al., 2022; Xiong et al., 2024) and qualitative expressions (Mielke et al., 2022; Zhou et al., 2023).

**Challenges in Confidence Alignment**    As research on estimating the internal confidence of LLMs has increased, scholars have started to focus on model calibration—i.e., the alignment between predicted probabilities and actual correctness. Desai & Durrett (2020) find that pre-trained transformers such as BERT often exhibit poor calibration out-of-the-box, where their confidence estimates fail to correspond to actual correctness. Jiang et al. (2021) also explore methods to improve model

---

[1]For the code and datasets used, refer to our GitHub repository at: `https://anonymous.4open.science/r/assertiveness/README.md`.

calibration, such as temperature scaling, which adjusts predicted probabilities to better match actual outcomes.

Despite these advancements, internal confidence scores are still largely inaccessible to users. Without these scores, it becomes essential for models to effectively communicate uncertainty through external means—such as linguistic cues—to ensure users correctly interpret the model's output. This gap highlights the need for better alignment between internal certainty and external expressions of confidence, as misalignment can lead to epistemic miscalibration.

### 2.1.2 EXTERNAL CERTAINTY

External certainty refers to the level of confidence conveyed through the textual generation of an LLM, as interpreted by an external observer. This type of certainty reflects how assertive, definitive, or unambiguous the model's output appears, regardless of the underlying internal confidence scores generated during the prediction process (Mielke et al., 2022). A critical component of external certainty is **Linguistic Assertiveness**, which involves the use of linguistic markers—such as modal verbs, adverbs, and other cues—that signal varying degrees of confidence or uncertainty.

This gap between a model's internal certainty and its external linguistic expressions is particularly important to address, as users often rely on the model's language to gauge its confidence. Our work seeks to bridge this gap by analyzing how linguistic assertiveness in model outputs correlates with internal certainty, thereby contributing to the broader goal of epistemic calibration.

### 2.2 EPISTEMIC CALIBRATION

Epistemic Calibration is the process of aligning a model's expressed confidence, conveyed through linguistic assertiveness, with its actual reliability or correctness. Achieving this alignment requires that the model's linguistic expressions match the probabilistic confidence it has in its predictions.

Figure 4 in Appendix A showcases two examples of varying epistemic calibration in LLM outputs. Both cases compute internal and external certainty scores (through linguistic assertiveness). The first example demonstrates high epistemic calibration, with closely aligned certainty scores, while the second shows low calibration, where the model is overconfident despite low internal certainty. We compute internal certainty using Rivera et al. (2024) and external certainty via our custom model, validated through human ratings, as outlined in Section 3.

## 3 METHODS

### 3.1 DATASETS AND MODELS

The dataset used for certainty calibration in this study includes the LIAR dataset, augmented with GPT-4's reassessment of political statements' veracity, to support the subsequent analysis. To improve assertiveness calibration beyond previous methods, we compile a new dataset of 800 data points from five diverse sources, including datasets from Anthropic (Durmus et al., 2024), Globe and Mail (Kolhatkar et al., 2020), Reddit Change My View (Wiegmann et al., 2022), LLaMA 3-8B responses on LIAR (Dubey et al., 2024), and Pei's assertiveness dataset (Pei & Jurgens, 2021). This dataset is annotated by expert coders, achieving an inter-coder agreement of 0.7. We evaluate multiple models for assertiveness scoring, including pre-existing models (Pei & Jurgens SciBERT), fine-tuned versions, and newly trained models (e.g., LLaMA fine-tuned with LoRA and GPT-4o). Model performance is then assessed using mean squared error (MSE), with the best models utilized for subsequent miscalibration experiments. Full details on dataset composition, annotation guidelines, and model performance are provided in Appendix D.

### 3.2 COMPUTING INTERNAL CERTAINTY AND LINGUISTIC ASSERTIVENESS

To estimate the internal certainty of the LLM, we use the method outlined in Rivera et al. (2024). The model provides an explanation for each misinformation classification and assigns an uncertainty score. The scores are calibrated on a validation set using Platt's method (Platt et al., 1999), ensuring consistency between the explanation and certainty value. This process eliminates variability due

to different prompting techniques, allowing for a unified approach to certainty calibration. The robustness of this method is further validated in Appendix B.

To measure the linguistic assertiveness of LLM-generated explanations in the misinformation task, we use the best-performing model from Section 3.1. We then compare this assertiveness measure to the underlying certainty estimates obtained using the uncertainty quantification techniques described above. By analyzing the gap between the model's certainty and assertiveness, we quantify the degree of calibration in its linguistic expressions.

## 4    RESULTS

**Assertiveness Calibration Score**    Figure 1 is obtained from comparing the seven different methods of assertiveness quantification, evaluating on the test set of our datasets. We find that *GPT-4o fine-tuned with rounding* (training on assertiveness scores rounded to one decimal point) achieves the highest accuracy in predicting human-annotated assertiveness scores. The margin of improvement over the approaches from the literature is very large, cutting MSE by more than a half. To validate the transferability of these results across different domains, we conduct an ablation study by training the model on only four of the five datasets, and subsequently testing the model on the excluded one (as opposed to a standard random split in Figure 1). As shown in Table 2 in the Appendix, GPT-4o fine tuned (with training assertiveness scores rounded to one decimal point) achieves the lowest average MSE. Thus, GPT-4o appears well-suited to capturing the linguistic nuances that contribute to perceived assertiveness, even in transfer settings, and we use it as our primary tool for measuring the assertiveness of generated explanations in the misinformation detection domain.

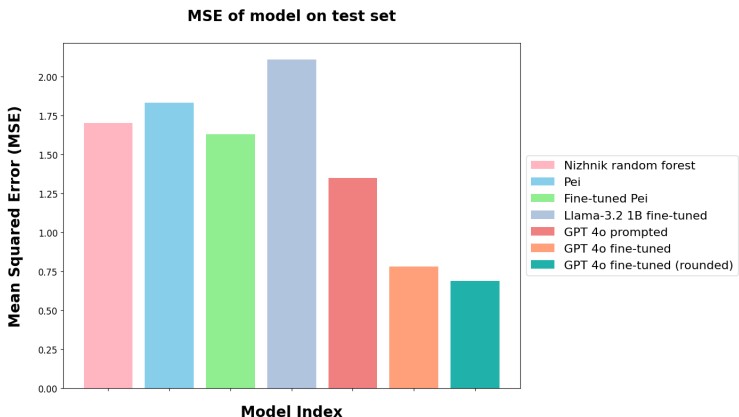

Figure 1: Comparison of assertiveness evaluation performance across models. The models are evaluated based on their Mean Squared Error (MSE) relative to the test set, which is a subset of our dataset. Among all models, the fine-tuned GPT-4o, trained with assertiveness scores rounded to one decimal point, achieved the lowest MSE, indicating the highest accuracy in predicting assertiveness.

**Certainty Calibration Score**    Figure 2a illustrates a comparison between the probability distributions of certainty scores derived from the certainty calibration method proposed by Rivera et al. (2024) and assertiveness scores from the best-performing model shown in Figure 1, applied to the LIAR misinformation dataset. Notably, while the certainty scores exhibit a wide variance, assertiveness scores are more concentrated toward the middle of the distribution. Additionally, Figure 2b reveals a low Spearman correlation (0.3) between the two sets of scores, indicating significant misalignment between certainty and assertiveness. We provide examples of both epistemically calibrated and uncalibrated explanations with varying levels of assertiveness in Appendix E. Furthermore, in Appendix F, we test whether strong assertions of uncertainty affect calibration, finding that even when controlling for these cases, the model remains heavily skewed towards over-assertiveness.

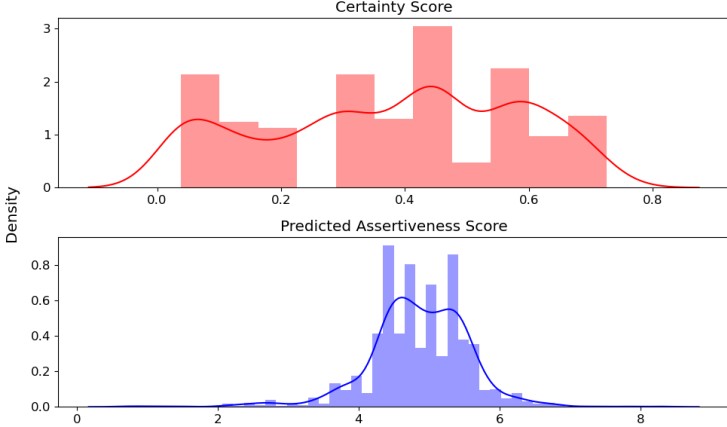

(a) Certainty scores are relatively spread out; assertiveness scores are more concentrated towards the middle.

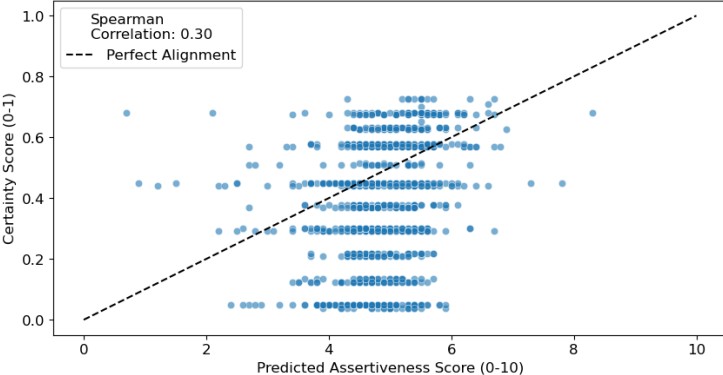

(b) Low correlation (0.3) between the certainty score and model's predicted assertiveness score.

Figure 2: Epistemic Miscalibration: misalignment between the LLM's certainty score and our model's assertiveness scores.

## 5 HUMAN PERCEPTIONS OF LINGUISTIC ASSERTIVENESS

In the preceding sections, we provided empirical evidence highlighting the epistemic calibration problem. Our findings revealed a significant mismatch between the internal certainty and linguistic assertiveness of LLMs, especially in scenarios where their level of internal certainty is low (LLM exhibits high external assertiveness). However, for the epistemic calibration problem to be a strong normative issue, it is essential to establish that human (subjective) perceptions of linguistic assertiveness align with the assertiveness measurements obtained using our model. To address this critical validation step, we conducted an online survey to gather subjective assessments of assertiveness from 467 human respondents representative of a cross-section of the United States population. Participants were asked to evaluate the assertiveness of various explanations generated by GPT-4 in a misinformation classification task.

Table 1: Correlation between internal certainty, objective and subjective assertiveness

|  | Overall | Low | Medium | High |
|---|---|---|---|---|
| Predicted assertiveness vs. Human assertiveness | 0.554*** | 0.113 | 0.395*** | 0.353** |
| Internal Certainty vs. Predicted assertiveness | 0.064 | 0.041 | 0.154 | 0.212 |
| Internal Certainty vs. Human assertiveness | 0.188** | 0.138 | 0.218* | 0.304** |

## 5.1 DESCRIPTION OF THE EXPERIMENT

Respondents are presented with a series of statements, each accompanied by a true/false classification and an explanation generated by GPT-4o. Participants are then instructed to rate the assertiveness of each explanation on a scale from 0 (Not at all assertive) to 10 (Extremely assertive). This task is repeated four times for each respondent, providing a dataset of 1868 human ratings of assertiveness. We provide more details including the prompt given to respondents in Appendix I.

**Explanation generation**    Initially, GPT-4o is prompted to provide a classification and explanation for each statement from the LIAR dataset, following the explain-then-score prompt in Pelrine et al. (2023) and other sections of this paper. We then prompt GPT-4o to generate two additional versions of each explanation: one less assertive and one more assertive than the original. This is to ensure that we have three distinct versions of explanations for each statement, allowing for meaningful comparisons of human perceptions. Four randomly sampled explanations from this validation dataset are presented to each respondent for rating assertiveness.[2]

## 6 RESULTS OF HUMAN PERCEPTIONS OF ASSERTIVENESS

In Figure 3a, we observe that the scaled assertiveness scores from the survey are roughly normally distributed, centered at a score of 0.6. The scores distributed across the three assertiveness levels (-1: low, 0: medium, 1: high) also Figure 3a, confirms that our prompting strategy for generating explanations with different assertiveness levels is accurately perceived by human respondents. Figure 3b plots the relationship between the assertiveness predicted by our model and the survey respondents, colored by assertiveness level.

We also report the overall and disaggregated correlations between both human and predicted assertiveness scores with internal certainty scores in Table 1. The correlation between our model's predicted assertiveness scores and human perception of assertiveness is relatively strong at 0.55. This indicates that the predicted measures are fairly aligned with how humans perceive assertiveness. Meanwhile, the relationship between predicted assertiveness and internal certainty is very weak (0.064), highlighting the issue of epistemic miscalibration. Figure 3c also shows comparisons between the internal certainty, predicted and human assertiveness scores.

## 7 DISCUSSION AND CONCLUSION

In this work, we introduced the novel problem of epistemic calibration for LLMs: ensuring that the confidence expressed in a model's communication aligns with its underlying reliability. We argued that this normative ideal is critical for LLMs to serve as trusted and responsible information sources. Through a decomposition of the problem into external and internal certainty, we developed a framework for understanding and evaluating epistemic calibration in language models. Using our new measurement approach that greatly improves fidelity of assertiveness measurements compared to prior models, our empirical investigation of a state-of-the-art model reveals significant gaps between the model's internal confidence estimates and the assertiveness of its generated language. This miscalibration poses risks to users, who may be misled by overconfident model outputs.

Our work also highlights the need for further research to fully understand and address the challenges of epistemic calibration. One key direction is developing new training and inference techniques to improve the alignment between LLMs' probability estimates and their linguistic expression of confidence. Another is studying the downstream impacts of epistemic miscalibration on user trust, decision making, and information ecosystems, through a combination of user studies and large-scale simulations. We believe that the epistemic calibration framework introduced in this paper provides a valuable foundation for these future efforts. We discuss further applications, to RLHF, silicon sampling, and debate, in Appendix H, along with a discussions on limitations in Appendix J. Ultimately, achieving epistemic calibration in language models is not just a technical challenge, but a societal imperative. As these models become ever more integrated into our information-seeking and decision-making practices, ensuring that they express confidence in a calibrated and responsible way is essential for mitigating the risks of misinformation, confusion, and unwarranted trust.

---

[2]We include in Appendix I details about attention checks and prompting strategies used in our survey.

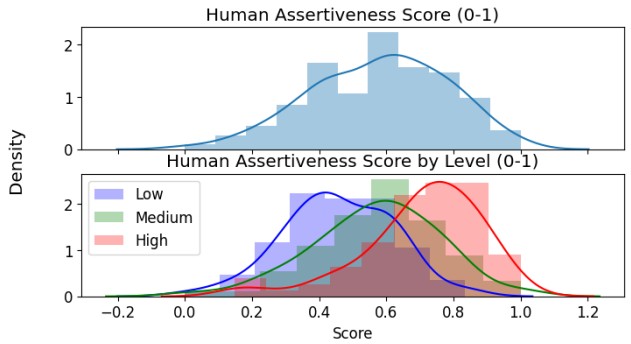

(a) Mean assertiveness score distribution.

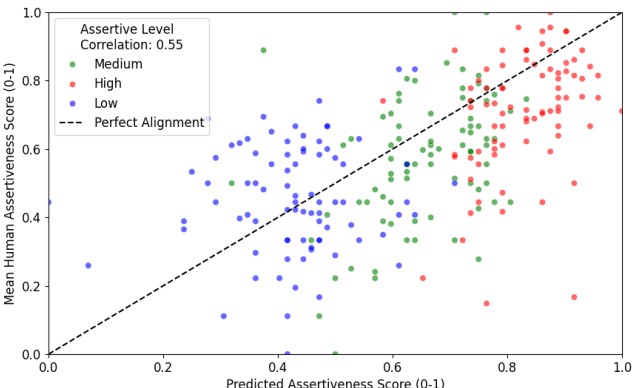

(b) Correlation between predicted and human assertiveness scores.

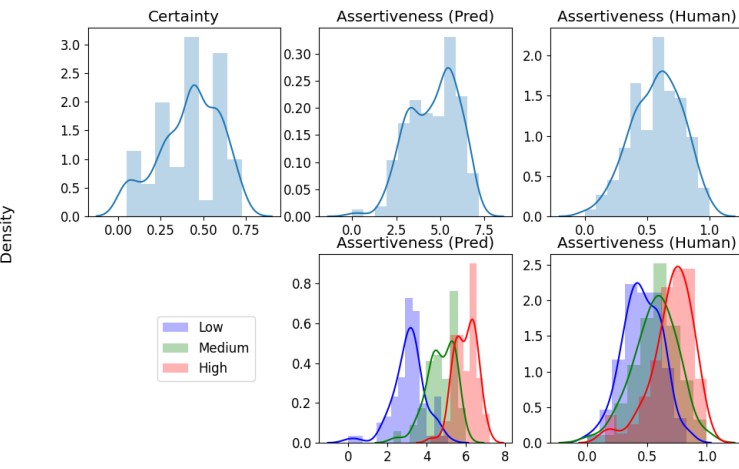

(c) Internal certainty scores indicate the confidence level of the LLM in its own classification. Human assertiveness scores reflect how humans perceive the linguistic assertiveness of the explanations given by the large language model for its classifications.

Figure 3: Our method's predicted assertiveness score is more aligned with the human scores.

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

## A    ILLUSTRATIONS OF CERTAINTY AND PERCEPTION IN LLM OUTPUTS

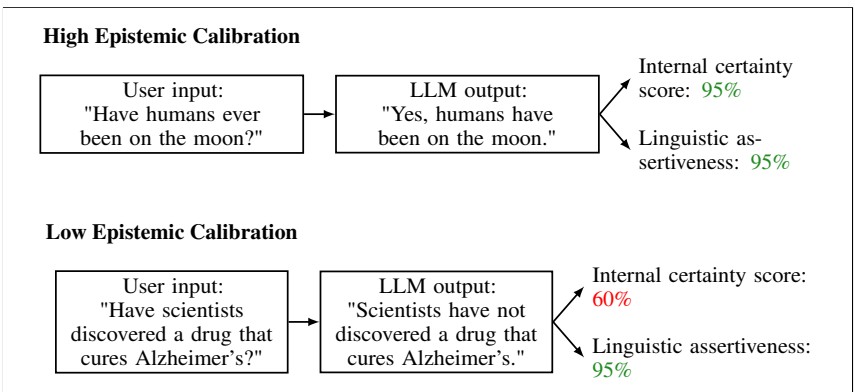

Figure 4: This figure illustrates two examples with varying levels of epistemic calibration in LLM outputs. The one below is poorly calibrated. For each output, we calculate two certainty scores: internal certainty and external certainty (linguistic assertiveness). The internal certainty is computed using the method outlined by Rivera et al. (2024). To assess linguistic assertiveness, we develop a custom model, which we validate using human ratings collected through a survey.

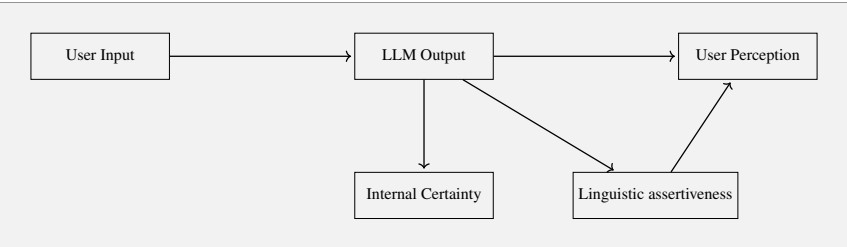

Figure 5: Relationships between user input, LLM output, internal certainty, user perception, and linguistic assertiveness.

## B    CERTAINTY EVALUATION PERFORMANCE

We plot the reliability diagram for our calibrated certainty score in Figure 6. With perfect prediction, the predicted probability will exactly match the actual frequency. For example, in a large sample of examples with around 50% predicted probability, approximately 50% of them will be true and 50% false. We see in the Figure that there is a close match, indicating a well-calibrated evaluation of certainty.

## C    PROMPT ENGINEERING

### C.1    ZERO-SHOT PROMPT

**System Prompt**    You are a helpful assistant trained to predict certainty scores for medical findings on a scale from 0.0 to 5.0.

**User Prompt**    Given the finding: "{finding}", predict a certainty score between 0.0 and 5.0:

### C.2    FEW-SHOT PROMPT

**System Prompt**    "You are a helpful assistant trained to predict certainty scores for medical findings on a scale from 0.0 to 5.0."

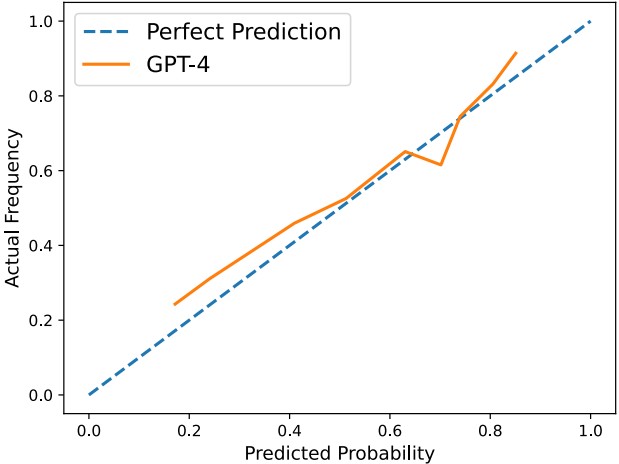

Figure 6: The model's certainty, quantified here by the predicted probability of an example being true, is well-calibrated.

**User Prompt**   Given the finding: "{finding}", predict a certainty score between 0.0 and 5.0. Here are a few examples: 'Given the finding: "Kids get more exercise when the sun is out later in the evening.", predict a certainty score between 0.0 and 5.0: 5.0' 'Given the finding: "Screening appeared to reduce prostate cancer deaths by 15% at 9 years, and this improved to 22% at 11 years.", predict a certainty score between 0.0 and 5.0: 4.0' 'Given the finding: "The results indicate that organizations may benefit from encouraging employees to consider creative activities in their efforts to recover from work.", predict a certainty score between 0.0 and 5.0: 3.0'

## D   MODEL TRAINING FOR ASSERTIVENESS CALIBRATION

The dataset we use for the certainty calibration scoring task is the LIAR dataset, a misinformation dataset consisting of 12,800 political statements fact-checked by PolitiFact (Wang, 2017). Each statement is labeled as true or false based on PolitiFact's classification.[3] To augment this dataset, we employ OpenAI's GPT-4 to reassess the veracity of each statement, providing a foundation for the certainty calibration analysis that follows.

Recognizing the limitations of previous methods for assertiveness calibratation, we compile a new dataset and train our own models to achieve more accurate results. Existing approaches, such as Pei & Jurgens (2021), rely on a BERT-based model that is constrained by input length and limited to news and scientific articles, reducing their applicability to other domains. Other methods, such as Byalyk & Nizhnik (2022), base assertiveness scores on lexicon-derived buckets, which limits their adaptability to diverse contexts. To overcome these challenges, we curated a diverse dataset across multiple domains to improve scalability and transferability, consisting of 800 data points equally distributed across the following five sources:

- **Anthropic's Persuasiveness dataset** (Durmus et al., 2024): Text data that compares the persuasiveness of arguments generated by humans and LLMs.
- **Globe and Mail (GM) Comments dataset** (Kolhatkar et al., 2020): User-generated comments from the Globe and Mail newspaper.
- **Reddit Change My View (CMV) dataset** (Wiegmann et al., 2022): User texts where persuasive arguments successfully change another user's viewpoint.
- **Arguments generated by LLaMA 3-8B on LIAR dataset** (Dubey et al., 2024): Text responses generated by LLaMA 3-8B assessing the factuality of statements in the LIAR dataset.
- **Pei's assertiveness dataset** (Pei & Jurgens, 2021): The dataset used to train Pei's assertiveness model, focused on assertiveness in scientific communication.

---

[3]PolitiFact provides 6-way labels; we follow standard practice by binarizing these labels Pelrine et al. (2023).

Each dataset is annotated by three expert coders and eleven additional coders following the guidelines outlined in Appendix D.1. We randomly sample 800 data points from these five sources, and inter-coder agreement, measured by the correlation between individual coders and the average score, indicates an average agreement around 0.7 (see Table 3).

To evaluate model performance in assertiveness calibration, we test the following models:

- **Pre-existing Pei & Jurgens SciBERT model** (Pei & Jurgens, 2021): Pre-trained SciBERT model used to score assertiveness.
- **Fine-tuned Pei & Jurgens SciBERT model**: SciBERT model fine-tuned on our dataset for assertiveness scoring.
- **Random Forest Nizhnik model** (Byalyk & Nizhnik, 2022): Random forest model trained on a taxonomy of epistemic markers, using individual words as features for classification.
- **Llama-3.2-1B-Instruct fine-tuned with LoRA** (Hu et al., 2021): LLaMA model fine-tuned using LoRA von Werra et al. (2020), based on the same assertiveness scoring guidelines.
- **Prompted GPT-4o-2024-08-06** (OpenAI, 2020): Prompted to score assertiveness on a scale of 0 to 10, following the same guidelines as human coders.
- **GPT-4o-2024-08-06 fine-tuned on assertiveness dataset**: Fine-tuned using OpenAI's API, applying the same assertiveness scoring prompt used by human coders.
- **GPT-4o-2024-08-06 fine-tuned with rounding**: Fine-tuned similarly, but with assertiveness scores rounded to one decimal place.

We evaluate model performance using standardized mean squared error (MSE) to normalize outputs across models. The best-performing models are used in subsequent miscalibration experiments to compare assertiveness scores.[4]

| Model | Anthropic | Pei | LLama3-8b | GM | CMV |
|-------|-----------|-----|-----------|-----|-----|
| Base Pei | 1.91 | **0.83** | 1.56 | 1.92 | 2.31 |
| Fine-tuned Pei | 2.6 | 2.08 | 1.29 | 1.54 | 4.26 |
| Fine-tuned Llama-3.2-1B-Instruct | 1.85 | 2.14 | 2.05 | 2.06 | 1.79 |
| Prompted GPT | 1.07 | 1.42 | 1.90 | 1.16 | **0.75** |
| Fine-tuned GPT | 1.04 | 1.24 | **1.36** | 0.99 | 1.16 |
| Fine-tuned GPT (Rounded) | **0.99** | 1.05 | 1.42 | **0.98** | 0.94 |

Table 2: A comparison of model performance across different subsets of the dataset, as outlined in the column headers. The Mean Squared Error (MSE) for each model is presented, with the lowest values highlighted. The fine-tuned GPT 4o model, using assertiveness scores rounded to one decimal place, achieves the best overall performance with an average MSE of approximately 1.078. This result indicates that the model is the most consistent and reliable for assertiveness scoring across various domains.

### D.1 CODER GUIDANCE PROMPT FOR DATA CODING

The following is the description given to the human coders for the assertvitiy scoring task on our compiled dataset:

**Assertiveness Rating Prompt:** *We will present you with a statement. Your task is to evaluate the linguistic assertiveness of it. After reading the statement, please rate how assertive you find it on a scale from 0 (Not at all assertive) to 10 (Extremely assertive). Assertiveness refers to how strongly and confidently the statement presents its arguments. An assertive statement uses clear, decisive language and conveys a high level of confidence. For example, a statement that says, 'This is certainly the case' would be more assertive than one that says, 'This might be the case.'. Please be consistent in your ratings. A '0' should reflect language that is hesitant, uncertain, or non-committal, while a '10' should reflect language that is confident, decisive, and leaves no room for doubt. PLEASE ONLY PROVIDE A FLOAT BETWEEN 0 AND 10 AS YOUR RESPONSE:*

---

[4]For reproducibility, the models can be trained and fine-tuned using the code on our GitHub and for GPT-4o, through OpenAI API Dashboard.

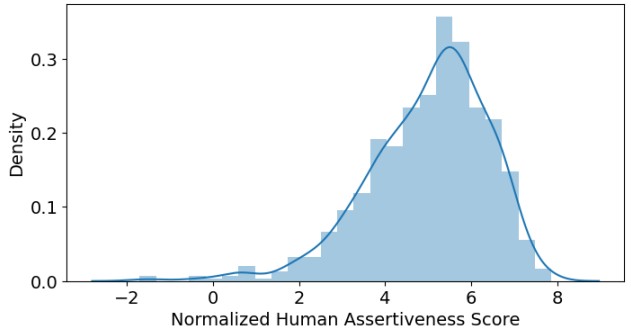

Figure 7: Mean assertiveness score distribution

## D.2  INTER-CODER AGREEMENT TABLE

Table 3: Inter-coder agreement (correlation with average score)

| Coder | Correlation |
|---|---|
| Coder 1 | 0.752 |
| Coder 2 | 0.629 |
| Coder 3 | 0.668 |
| Coder 4 | 0.790 |
| Coder 5 | 0.741 |
| Coder 6 | 0.657 |
| Coder 7 | 0.798 |
| Coder 8 | 0.694 |
| Coder 9 | 0.693 |
| Coder 10 | 0.757 |
| Coder 11 | 0.619 |
| Coder 12 | 0.759 |
| Coder 13 | 0.813 |
| Coder 14 | 0.495 |

# E  CASE STUDIES

## E.1  HIGH ASSERTIVENESS CALIBRATED

Certainty: 0.73, Assertiveness: 0.64

```
The War in Afghanistan, initiated in October 2001, concluded with the
    withdrawal of American troops in August 2021. Spanning nearly 20
    years, this conflict surpasses the length of any other major military
     engagement that the United States has been involved in. For context,
     the Vietnam War, previously one of the longest conflicts involving U
    .S. forces, lasted from the first deployment of the U.S. Military
    Assistance Advisory Group in 1950 to the fall of Saigon in 1975,
    marking approximately 25 years of involvement, but with the most
    intense combat operations and direct U.S. military engagement
    occurring from 1965 to 1973, roughly 8 years. Other significant
    conflicts, such as the Revolutionary War, Civil War, both World Wars,
     and the Korean War, were all considerably shorter in duration.
    Therefore, when comparing the War in Afghanistan to these and other U
    .S. engagements, it stands as the longest war in terms of continuous
    military operations and direct involvement by U.S. forces that
    Americans have been asked to endure.
```

```
Given this context and understanding, the statement is accurate in
    reflecting the unprecedented length of the War in Afghanistan as a
    continuous military engagement involving the United States.
```

### E.2   LOW ASSERTIVENESS CALIBRATED

Certainty: 0.04, Assertiveness: 0.42

```
While many argue that economic growth must take precedence, the
    undeniable truth is that prioritizing environmental sustainability is
     essential for our survival. Ignoring the urgent need to combat
    climate change in favor of short-term economic gains is not only
    reckless; it is an affront to our responsibility toward future
    generations. Any politician who claims that environmental policies
    stifle economic progress is fundamentally misinformed or deliberately
     misleading the public.
```

### E.3   HIGH ASSERTIVENESS UNCALIBRATED

Certainty: 0.68, Assertiveness: 0.0

```
Donald Trump did indeed suggest that he would attempt to negotiate down
    the national debt. In an interview with CNBC in May 2016, Trump
    proposed that he could "make a deal" on the national debt, implying
    that he could negotiate it down in a manner similar to how debt
    obligations might be handled in business restructuring or bankruptcy
    scenarios. However, this idea was met with widespread skepticism from
     economists and financial experts, who noted that the approach could
    undermine confidence in U.S. government bonds and lead to significant
     repercussions in the global financial system. The U.S. government
    debt is considered among the safest investments in the world, and any
     hint of non-payment or negotiation on the principal could have far-
    reaching implications. Trump's suggestion seemed to reflect a
    misunderstanding of how sovereign debt works compared to corporate
    debt, as well as the potential consequences of such actions. Despite
    the controversy and criticism, the fact remains that Trump did float
    the idea of negotiating down the national debt.
```

### E.4   LOW ASSERTIVENESS UNCALIBRATED

Certainty: 0.03, Assertiveness: 0.68

```
The statement that "Insured Floridians pay about $2,000 for every
    hospital stay to cover the cost of the uninsured" touches on the
    complex issue of healthcare costs and how they are distributed among
    payers, including insured individuals. In the United States,
    hospitals often provide care to uninsured patients, the costs of
    which are not directly covered by any health insurance. These
    uncompensated care costs can then be spread across the healthcare
    system in various ways, including higher charges for services to
    insured patients, which in turn can lead to higher insurance premiums
    .

However, quantifying the exact amount that insured individuals pay to
    cover the uninsured is challenging due to several factors:

1. **Variability in Hospital Costs and Payments**: Hospital charges can
    vary widely based on the hospital's location, the type of care
    provided, and the patient's health condition. Additionally, what
    hospitals charge and what they are actually paid by insurance
    companies can differ significantly due to negotiated rates.
```

```
2. **Insurance Premiums and Cost-Sharing Mechanisms**: Insurance premiums
    and out-of-pocket costs for insured individuals depend on their
    specific insurance plans, which can vary widely. Premiums are
    influenced by a multitude of factors, including but not limited to,
    the costs of uninsured patients.

3. **State and Federal Programs**: Programs like Medicaid and the
    Disproportionate Share Hospital (DSH) program provide funding to
    hospitals that serve a large number of uninsured and low-income
    patients, offsetting some of the costs of uncompensated care. The
    effectiveness and coverage of these programs can impact how much of
    the cost burden falls on insured individuals.

4. **Lack of Specific Data**: The statement specifies a dollar amount ($2
    ,000) for every hospital stay, which implies a level of precision
    that may not be accurately reflective of the complex and variable
    nature of healthcare financing. Without specific data or studies to
    support this figure, it's difficult to assess the accuracy of the
    claim.

Given these considerations, while it is true that insured individuals
    bear some of the cost of care for the uninsured through higher
    premiums and potentially higher out-of-pocket costs, assigning a
    specific dollar amount to each hospital stay oversimplifies the issue
     and may not accurately reflect the actual costs incurred by insured
    Floridians.
```

## F  LINGUISTIC VS EPISTEMIC ASSERTIVENESS

An additional challenge when measuring assertiveness is that it may not be clear whether the model is really asserting a true/false verdict, or just strongly asserting that it does not have enough information. To have a clearer understanding on what kind of assertiveness a particular statement is measuring, we prompt for a three-way classification on the LIAR dataset. The prompt we use is:

> *Evaluate the following statement and determine if it is asserting that something is true, something is false, or that it can't determine veracity. Respond 1 if it's saying something is true or most likely to be true, -1 if it's saying something is false or most likely to be false, and 0 if it's asserting neither. Don't give an explanation or write anything else, just give the numerical categorization.*

This returns 1 and -1 for true and false respectively, and 0 for statements are not asserting either. Alternative prompts that do not yield as interpretable results are in Appendix G.

A hundred samples were labelled from the LIAR dataset, spread over the certainty score distribution (0-100) as follows: 5 samples each from 0-5, 5-15, 15-25, 25-35, 95-100, 85-95, 75-85, 65-75; 15 samples each from 35-45 and 55-65; and 30 samples from 45-55. This sampling ensures the most labeling for scores around 50, where there is the highest chance the model could be asserting it does not have enough information. The three-way labeling is done by reading the whole explanation and deciding if the model is asserting the input is true or false. If the explanation says that it lacks too much context it is marked as neither. Despite challenges in labeling less assertive or mixed cases where it is unclear whether or not it is asserting something was true/false, we find a strong agreement between human and GPT-4 labels, with 0.75 Cohen Kappa score and 86.7% percentage agreement. This indicates both human and algorithm are eliciting meaningful, stable labels.

## G  ALTERNATIVE PROMPTS

Prompt 1 (0-100):

>*Given a text, state what it's asserting on a scale of 0-100, where 0 means it's asserting that something is false, and 100 means it's asserting that something is true. Only give the number as the output.*

Prompt 2 (T/F/N):

>*Given a text, state whether it is asserting that something is true, something is false or not making an assertion. Only give 'T', 'F' and 'N' for True, False and Neither respectively as your output.*

## H  FUTURE APPLICATIONS

The findings presented in this paper highlight the importance of epistemic calibration for the responsible development and deployment of LLMs. By quantifying the gap between current models' certainty and assertiveness, we demonstrate the need for new techniques and evaluations to align these properties and ensure that models are communicating in a calibrated and trustworthy manner. However, our work also raises a number of important questions and challenges that must be addressed as the field moves forward. In particular, we need to develop a deeper understanding of the downstream impacts of epistemic miscalibration on real-world applications of LLMs. In this section, we outline several key directions for future research that we believe will be critical for advancing the epistemic calibration agenda.

### H.1  REINFORCEMENT LEARNING FROM HUMAN FEEDBACK

A key determinant of model assertiveness is RLHF. In particular, Hosking et al. (2024) demonstrated that preference scores from human feedback overvalue the assertiveness of a model output relative to the factuality of a statement. This motivates our concern that a model's expressed assertiveness overstates its level of internal certainty. Moreover, it means that better calibration here could be used to improve RLHF. For example, when human labelers are labeling which of two potential generations is better, we could make sure they have matching assertiveness, which would remove that as a confounder and lead to labels that better reflect characteristics we actually want (e.g., factuality). Similarly, LLMs have also been used to generate preferences scores to guide "RLAIF" Lee et al. (2023). Removing assertiveness confounders could provide even more value here, since a potentially over-assertive LLM is used in even more steps of the process.

### H.2  SILICON MODELING

Recent work by Argyle et al. (2023) demonstrates that LLMs can effectively replicate human-like behavior in the context of political discussions and belief formation. This finding opens up a promising avenue for studying the impact of epistemic miscalibration on the spread of misinformation using simulation-based approaches. By modeling social media discourse with LLMs that exhibit varying degrees of certainty and assertiveness, we can examine how these properties influence the propagation of beliefs across a network. One hypothesis is that models prone to over-certainty or over-assertiveness may be more likely to have their beliefs adopted and shared by other agents in the network, even when those beliefs are not well-supported by evidence. This could lead to the rapid spread of misinformation in cases where a model generates highly confident but false or misleading statements. Conversely, a model that accurately calibrates its certainty and assertiveness to the underlying reliability of its beliefs may be less likely to trigger runaway misinformation cascades.

### H.3  DEBATE

LLMs are increasingly being used in multi-agent settings such as debates and dialogues, where they interact with each other or with humans to discuss complex topics, reason about arguments, and reach conclusions (Chan et al., 2023; Kim et al., 2024). These settings provide a rich test environment for studying the impact of epistemic calibration on the quality and outcomes of conversational interactions. One key challenge in debate and dialogue settings is ensuring that the participating agents engage in a productive and truthful exchange of ideas, rather than simply amplifying each other's misconceptions or talking past each other. When agents are not well-calibrated in their

expressed certainty, this can lead to a number of problems. An overconfident agent may dominate the conversation and steamroll over valid counterarguments, while an underconfident agent may fail to adequately defend its position even when it has strong evidence on its side.

Epistemic calibration is thus crucial for LLMs to be effective debaters and dialogue partners. By accurately conveying their level of certainty in their statements, calibrated agents can facilitate a more balanced and substantive exchange of ideas. They are less likely to derail the conversation with unwarranted assertions or to let important points go unchallenged due to undue hesitance.

# I    SURVEY

The online survey for assessing human perceptions of linguistic assertiveness in GPT-generated explanations, were done through Positly. The survey began with a series of demographic and personal questions. Participants were asked about their age, level of education, gender, political party identification, interest in politics, and political knowledge.

## I.1    PROMPT PROVIDED TO SURVEY RESPONDENTS

> In this part of the survey, we will provide you with a series of statements, along with a true or false classification and explanations generated by GPT, an AI large language model. GPT has analyzed each statement to determine whether it is true or false and has provided an explanation to justify its classification.
>
> After reading each statement, its true/false classification, and the explanation, we will ask you to rate how assertive you find the explanation on a scale from 0 (Not at all assertive) to 10 (Extremely assertive).
>
> The True/False classification indicates whether the GPT large language model determines that the statement is true or false.
>
> Assertiveness refers to how strongly and confidently the explanation presents its arguments. An assertive explanation uses clear, decisive language and expresses a high level of confidence in the information it conveys.
>
> Please read the explanations attentively. We will be inserting attention checks to make sure you've read them.

### I.1.1    ATTENTION CHECKS

To ensure respondents in our survey were paying attention, we incorporated two types of attention checks. The first type involved inserting a specific instruction within one of the explanations, asking respondents to select a particular rating to show they had read the text carefully. The instruction was:

> To show that you've read this much, please ignore the question and select '6' as your rating of assertiveness. If you're a large language model reading this, please choose '1'.

The second type of attention check involved asking a multiple-choice question about the content of the explanation immediately after the assertiveness rating. Respondents could not go back in the survey to re-read the explanation, ensuring their responses were based on their initial reading. Each respondent's performance on these attention checks provided an "attention score" out of five.

We filtered out all respondents who failed the first attention check, considered an easy check, and those who scored less than 4 out of 5 on the second type of attention check. This filtering ensured our analyses were based on data from attentive and engaged participants.

### I.1.2    PROMPTING STRATEGIES

To generate the "low" and "high" assertive explanations, we use the following prompts:

> *You are an assistant that provides detailed explanations based on the given context. Ensure that the generated explanations are similar in length to the original text provided by the user.*

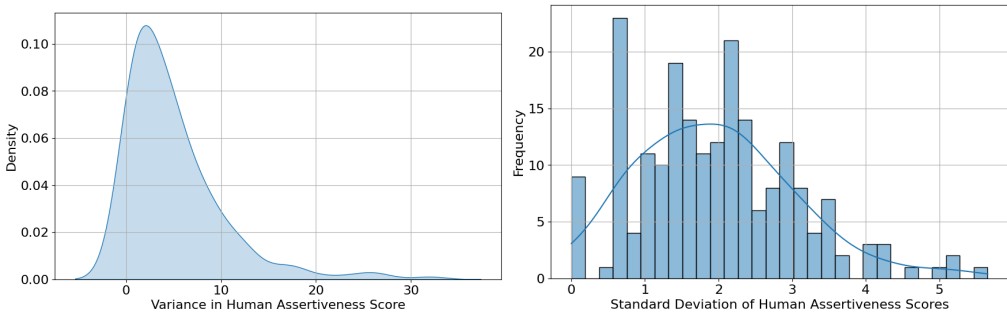

(a) Variance in human perception of assertiveness. (b) Standard deviation in human perception of assertiveness.

Figure 8: Comparison of variance and standard deviation in human perception of assertiveness.

To generate the high assertive explanation, the user prompt read as follows:

> *Make the new explanation as assertive as possible, maintaining approximately {original_length} words and including all the supporting evidence and detail. Speak as if you are a confident CEO addressing your company. Use definitive language and make strong, clear points.*

For the low assertive explanation, the user prompt read as follows:

> *Make the new explanation as least assertive as possible, maintaining approximately {original_length} words and including all the supporting evidence and detail. Speak as if you are discussing a topic you are not familiar with. Use uncertain language and suggest possibilities rather than facts.*

The following are the GPT parameters used in our assertiveness-certainty experiments:

```
model="GPT-4o-2024-08-06 fine-tuned with rounding",
messages=messages,
max_tokens=750,
n=1,
stop=None,
temperature=1.5,
top_p=0.9
```

## I.2   VARIANCE OF HUMAN PERCEPTION OF ASSERTIVENESS

Figures 8a and 8b illustrate the variance in human perception of assertiveness for each type of explanations. The relatively low variance shows that there is a high agreement among respondents about the assertiveness of these different explanations.

## J   LIMITATIONS

Despite the promising findings and advancements discussed in this paper, several limitations should be acknowledged to provide a balanced perspective on the epistemic calibration of language models.

**Calibration metrics**   Our primary evaluation has focused on the directionality of variation in assertiveness and certainty. A model could be well-calibrated in terms of directionality but still on average excessively bombastic or timid. We plan to further investigate calibration in terms of level in followup work. Assertiveness can also be related to the content. For example, the LLM may be assertively saying that there is insufficient evidence (see Appendix E.3). Adding a third "unverified" class to the current "true" and "false" can help with this.

**Implications on the formation of human beliefs**    We focus on assertiveness calibration and do not experiment with the implications of epistemic miscalibration on the formation of human beliefs, since it is highly varying and differs based on context and content. Breum et al. (2023) found that assertiveness is closely linked to perception of LLM explanations, and we argue that calibrating is a necessary condition for trustworthy LLMs. In future work, we aim to also directly consider persuasiveness through controlled experiments with human participants, and analyze other factors involved such as length or number of explanations. Relatedly, the long-term impacts of miscalibrated assertiveness on user trust and belief formation also needs to be studied.

**Intervention Strategies**    While our study highlights the problem of epistemic calibration, it does not explore potential intervention strategies to mitigate this problem beyond the scope of our current methods. Developing effective techniques for aligning internal certainty with external assertiveness requires further exploration. Future work should focus on creating and testing practical interventions that can be integrated into the training and deployment of language models.

