# OpenReview forum: "Epistemic Integrity in Large Language Models"
_NeurIPS.cc/2024/Workshop/SafeGenAi — SafeGenAi Poster_

### Official Review · Reviewer_cz6z · 2024-10-09
**Comments on "Epistemic Integrity in Large Language Models"**

**Rating:** 7
**Confidence:** 4

**Review:**

This study tackles the issue of miscalibration between LLMs' internal certainty and assertiveness in their language outputs. The authors introduce a human-labeled dataset and a novel method to measure the mismatch, revealing a significant misalignment between a model’s actual confidence and its presented output.

Strengths:
1. The research question is particularly compelling. While prior studies have examined internal and external inconsistencies in LLMs, this work provides a fresh perspective by focusing on the gap between the model’s internal certainty and its linguistic assertiveness. This contribution is especially valuable given its implications for improving LLM regulation and safety, addressing a critical need in the field.
2. The study is well-executed, with the authors demonstrating the robustness of their findings across multiple datasets. Furthermore, they bridge model predictions with human perception through an online survey, adding an important human-centered dimension to the analysis. This strengthens the relevance and applicability of the findings.

Limitations:
1. The study’s focus is confined to misinformation within political statements. While this context is important, it limits the generalizability of the findings. Broadening the scope to other domains—such as medical, legal, or scientific statements—would provide a more comprehensive understanding of miscalibration in LLMs.
2. In Table 1, the reported correlation between predicted assertiveness and human-perceived assertiveness is 0.5 (presumably Pearson correlation). The authors describe this correlation as “relatively strong,” but the criteria for what constitutes “strong” are not well-defined. It would be helpful for the authors to clarify the standards they are using to interpret these statistical relationships, particularly in the context of similar studies.
3. While the study quantitatively establishes the presence of miscalibration, it would benefit from a qualitative analysis of specific cases. Understanding the types of contexts in which misalignment occurs would provide deeper insight into the nature of the problem. Furthermore, exploring whether these misalignments can be predicted based on the prompting context could enhance the study’s practical implications for model design and deployment.

---

### Official Review · Reviewer_KBqE · 2024-10-09

**Rating:** 7
**Confidence:** 3

**Review:**

**Short Summary:** This paper introduces the problem of epistemic calibration, where there is a difference between the certainty of the answer produced by an LLM and the model’s internal certainty. Through training a new model for measuring LLM external certainty and a human survey, they show that both (1) current LLMs have a significant difference between internal and external certainty and (2) the certainty of LLM’s answer affects how people perceive the results.

**Quality and Clarity:** While the content in each section is straightforward, the main paper is too dependent on the appendix. The figures were not integrated into the paper, making it challenging to read the overall paper.

**Originality:** The epistemic calibration problem introduced in this paper is novel.

**Strengths:**
- The paper makes a strong argument on why LLMs not having epistemic calibration is a problem when introducing the problem and in their human survey.
- The GPT-4o model fine-tuned in this paper is well documented, and a convincing analysis shows that it effectively measures assertiveness.
- The experiments run seem to be sound and the resulting figures are strong evidence supporting the claims made in the paper.

**Weaknesses:**
- Correlation scores are missing the p-values, making it unclear whether the correlation is statistically significant.
- The paper discusses multiple approaches to calculating the internal certainty score but only seems to evaluate Rivera et al. It is unclear why this approach for calculating the score was picked or why other approaches discussed in Section 2.1.1 were not evaluated.
- The analysis of the later figures in the paper could be better connected to the points being addressed.

**General Review:** While the current version of the paper has writing problems, the problem discussed in this paper is novel and relevant to safe generative AI. The other weaknesses are not critical problems. The approach and results are sound and clearly show the importance of epistemic calibration.